# Karyotypes and *COI* Gene Sequences of *Chironomus* sp. Le1 (Kiknadze and Salova, 1996), *Ch. laetus* (Belyanina and Filinkova, 1996) and Their Hybrid from the Yamal Peninsula, Arctic Zone of Russia

**DOI:** 10.3390/insects13121112

**Published:** 2022-11-30

**Authors:** Viktor Bolshakov, Alexander Prokin, Dmitry Pavlov, Azamat Akkizov, Ekaterina Movergoz

**Affiliations:** 1Papanin Institute for Biology of Inland Waters Russian Academy of Sciences, Yaroslavl reg., Nekouz prov., Borok 152742, Russia; 2Ecological-Analytical Laboratory, Cherepovets State University, Lunacharski 5, Cherepovets 162600, Russia; 3Department of Biology, Geoecology and Molecular Genetic Foundations of Living Systems, Kabardino-Balkarian State University Named after H. M. Berbekov, Chernyshevsky St., 173, Nalchik 360004, Russia

**Keywords:** karyotype, hybrid, Chironomus, *Chironomus laetus*, *Ch.* sp. Le1, COI, DNA-barcoding, comparative cytogenetics

## Abstract

**Simple Summary:**

Chironomids, or non-biting midges, are one of the most abundant insect families. Their larvae are an important component of various aquatic ecosystems. In the *Chironomus* genus, sibling species are found that are not distinguished by morphology. Due to the presence of giant polytene chromosomes in their salivary gland, it is more convenient to use cytogenetics for species identification. The molecular genetic method also has limitations and depends on the level of intra- and interspecific variability of species. We used morphological, cytogenetic, and molecular genetic approaches to the study of chironomid larvae from two lakes on the Yamal Peninsula. We found a few larvae, which by morphology were very similar to the *Chironomus riihimakiensis* group. By cytogenetics, we identified the species as *Ch. laetus*, the first species with arm combinations AE BC DF G and propose the name of a new cytocomplex—“laetus”. We also found a hybrid, *Ch. laetus* × *Ch.* sp. Le1, which is the first hybrid between species from different cytocomplexes. Molecular-genetic analysis of *COI* gene sequences has shown high values of genetic distance between *Ch. laetus* and species from the *Ch. riihimakiensis* group. Molecular genetic data suggests the presence of a subgroup with *Ch. laetus*.

**Abstract:**

The study of the biological diversity of the Arctic zone yields intriguing results. Initial research on the lakes of the Yamal Peninsula resulted in the identification of *Chironomus laetus* and the hybrid *Ch. laetus* × *Ch*. sp. Le1. To avoid misidentification, we used morphological, cytogenetic, and molecular genetic approaches. By cytogenetics, in *Ch*. sp. Le1, seven banding sequences were found: Le1A1, Le1B1, Le1C1, Le1D1, Le1E1, Le1F1, and Le1G1. The karyotype of *Ch. laetus* was mapped for the first time; it is the first species with the arm combinations AE BC DF G. We propose the name of a new cytocomplex—“laetus”. DNA-barcoding of the *COI* gene was carried out for *Ch. laetus* and *Ch. laetus* × *Ch.* sp. Le1 for the first time. The estimated genetic distance between the sequences of *Ch. laetus* and *Ch. riihimakiensis* is 2.3–2.5%. The high similarity in morphology, banding sequences, and the possibility of hybridization indicate a close relationship between *Ch. laetus* and *Ch*. sp. Le1, which is assumed to be the northern variant of *Ch. riihimakiensis*. Molecular genetic data suggests the presence of a subgroup with *Ch. laetus*.

## 1. Introduction

The Arctic region has recently attracted the attention of biodiversity researchers. Special attention is dedicated to studying a species-rich family of chironomids, which play an important role in aquatic ecosystems.

Only a few species have been previously recorded from the region of investigation: *Chironomus beljaninae* (Wülker, 1991) [1], *Ch. plumosus* (Linnaeus, 1758), *Ch. nigrifrons* (Linevich and Erbaeva, 1971) [2], and also a lot of unidentified Chironomus spp., which were described only on the basis of larval morphology [3]. The vicinities of Vorkuta city (Figure 1), about 400 km from the studied location, is a type locality for three species: *Chironomus laetus* (Belyanina and Filinkova, 1996), *Ch. frequentatus* (Belyanina and Filinkova, 1996), and *Ch. borealis* (Belyanina and Filinkova, 1996), described without details of their karyotypes [4].

The main issue in the studies of *Chironomus* is the incorrect identification of species by morphological characteristics, especially at the larval stage, when reliable identification of some sibling species can be completely impossible. The most cost-effective and fastest method for the analysis of thousands and thousands of individuals in a short period is barcoding, which uses known sequences of DNA in any stage of the organism’s development. Recently, this method was used to study the biodiversity of invertebrates in Canada [5], Norway [6], and South Korea [7]. The main disadvantage of this method is its low accuracy, which is associated with misidentification of species whose sequences have been deposited in genetic information databases (GenBank and BOLD) [8,9]. 

For example, for all Chironomidae registered in Germany, about 65% of the sequences do not have a species level of identification, the so-called “dark taxa” [10]. Previously, we found that the species name “*Chironomus melanotus*” matches several *COI* gene sequences, and the genetic p-distance between different populations varied from 0.4 to 12.4%, indicating that not all sequences belong to *Ch. melanotus* [11]. As we know, the genetic distance threshold is not suitable for all *Chironomus* species and depends on intra and interspecific sequence divergences; for the *COI* gene, it varies from 9 to 20% and, in rare cases, from 1 to 4% [12]. Due to the lack of sufficient information, we will use the 3% threshold suggested in the work [12,13,14,15,16]. 

The most accurate method of *Chironomus* identification is karyological analysis, which uses well-studied chromosomal banding sequences [17,18]. Many new cryptic species were described using this approach [18]. 

During analysis of the karyotypes of larvae from the Yamal Peninsula, we found a very strange chromosomal arm combination: “AB AE BC CD DF EF G”, which consists of two different cytocomplexes. After thorough analysis, we concluded that it was an interspecies hybrid. One of the parents is *Ch.* sp. Le1, which belongs to the “thummi” cytocomplex [17], previously known from lakes in Yakutia [19].

*Ch.* sp. Le1 is a member of the *Chironomus riihimakiensis* (Wülker, 1973) group, which also includes *Ch. tuvanicus* (Kiknadze, Siirin and Wülker, 1992), *Chironomus* sp. Tu1 (Kiknadze, Siirin and Wülker, 1993), and *Chironomus* sp. Al1 (Kiknadze, Siirin and Wülker, 1992) [20]. *Ch.* sp. Le1 was recorded from the Lena River delta [19], the Novaya Zemlya Islands, Dikson (Taimyr Peninsula), Alaska and Ellesmere in North America [17], and Wrangel Island [21], which suggests that the species has a circumpolar distribution [21,22].

Another parent is the *Ch. laetus* (Belyanina and Filinkova, 1996). Belyanina and Filinkova (1996) noted that *Ch. laetus* belong to the Chironomus “pseudothummi” cytocomplex, with the chromosomal arm combination AE BF CD G. However, analysis of its karyotype published in the aforementioned study showed that the authors have incorrectly identified chromosomal arms and the true combination was AE BC DF G; an unknown (new) cytocomplex. 

Finding *Chironomus* hybrids in nature occurs rarely. Only a few cases of natural hybrids were recorded: *Ch. plumosus* × *Ch. entis* (Shobanov, 1989); *Ch. plumosus* × *Ch. muratensis* (Ryser, Scholl and Wülker, 1983) [23]; and *Ch. riparius* (Meigen, 1804) × *Ch. piger* (Strenzke, 1959) [24,25]. It is noted that hybridization occurs between a few members of the *Chironomus oppositus* (Walker, 1856) group [26]. Several hybrids were also obtained under laboratory conditions: *Ch. plumosus* × *Ch. vancouveri* (Michailova and Fischer, 1986); *Ch. plumosus* × *Ch. balatonicus* (Dévai, Wülker and Scholl, 1983) [27]; *Camptochironomus tentans* (Fabricius, 1805) × *C. pallidivittatus* (Malloch, 1915) [28,29]; and *C. tentans* × *C. dilutus* (Shobanov, Kiknadze and Butler, 1999) (our data). Hybrids between species that belong to different cytocomplexes have not been reported previously.

In this work, we use a complex approach to the species identification of *Chironomus*. Preliminary investigation has shown the presence in GenBank and BOLD of about 50 unidentified *COI* gene sequences with high similarity to *Ch. laetus*, from Canada and Norway, and most of them situated outside of the Arctic Circle. The genetics of *Ch. riihimakiensis* have been studied insufficiently; only two *COI* gene sequences have been deposited in GenBank so far.

Chironomids are a very important component of aquatic ecosystems, and interest in them remains strong [7]. Molecular-genetic approaches are very useful in identifying the species at different stages of development, but this requires high accuracy in the first identification of species. The study’s goal is to learn more about the Arctic distribution, morphological, cytological, and molecular genetic characteristics of *Ch. laetus* and the hybrid of *Ch.* sp. Le1 × *Ch. laetus*.

## 2. Materials and Methods

Fourth-instar larvae of *Ch. laetus* and the hybrid of *Ch. laetus* × *Ch.* sp. Le1 were collected from two lakes in the northeast Yamal Peninsula. Five larvae were found in Lake Pidarmato (71.253612, 71.638747) in June 2021. The depth of the lake is 3–4 m, the bottom sediments are silted sand, the pH is 6.8, and the dissolved oxygen concentration is 9.49 mg/L. One more larva was found in an unnamed lake (71.120760, 72.270375) in August 2021. The depth of the lake was 1–2 m; the bottom sediments are silted sand with inclusions of organic matter (rotting moss); a pH of 6.4; and dissolved oxygen concentration of 9.16 mg/L.

The head capsules of all larvae were mounted on a slide in the Fora-Berlese solution (Figure 2). The morphological terminology proposed by Sæther [30] was used.

Karyotype analysis was performed on all larvae by the aceto-orcein technique [18]. A light microscope (Micromed-6C. LOMO, St. Petersburg reg., St. Petersburg, Russia) with a standard oil objective of ×100 and a digital camera (ToupCam5.1., Hangzhou, China) were used for microscopy analysis. For identification of chromosome banding sequences, we used the cytoMAPS [4,17,19,20,31,32]. All preparations of *Ch. laetus* and *Ch.* sp. Le1 have been deposited in the collection of the Papanin Institute for Biology of Inland Waters, Russian Academy of Sciences (Borok, Russia).

Three larvae from the Pidarmato Lake and one from an unnamed lake, which were studied karyologically and morphologically, were used for the DNA extraction using the «M-sorb-OOM» (Sintol, Moscow region, Moscow, Russia) kit with magnet particles according to the manufacturer’s protocol. For amplification of *COI* (cytochrome oxidase subunit I), primers LCO1490 (5′-GGTCAACAAATCATAAAGATATTGG-3′) and HCO2198 (5′-TAAACTTCAGGGTGACCAAAAAATCA-3′) were used (Eurogen, Moscow region, Moscow, Russia) [33]. The amplification reaction was carried out in a 25 μL reaction mixture (1× buffer, 1.5 μM MgCl_2_, 0.5 mM of each primer, 0.2 μM dNTP of each nucleotide, 17.55 μL of deionized water, 1 μL of template DNA, and 1 unit of Taq-polymerase (Evrogen, Moscow, Russia). PCR was performed at 94 °C for 3 min, followed by 30 cycles at 94 °C for 15 s, 50 °C for 45 s, 72 °C for 60 s, and a final cycle at 72 °C for 8 min. For the visualization of PCR products, 1% agarose gel was used, followed by purification with ethanol and ammonium acetate (3 M). Both strands were sequenced on an Applied Biosystems 3500 DNA sequencer (Thermo Scientific, Waltham, MA, USA) following the manufacturer’s instructions.

For alignment of *COI* nucleotide sequences, we used MUSCLE in the MEGA6 software [34]. To calculate pairwise genetic distances using Kimura 2-parameter (K2P) with codon position preferences: first, second, third, and noncoding sites; the MEGA6 software (Pennsylvania State University, State College, PA, USA) was used [35]. The program MrBayes v.3.2.6 was used for the Bayesian analysis [36,37] with previously suggested settings [15,38] for 1,000,000 iterations and 1000 iterations of burn-in, with nst = 6 (GTR + I + G). The phylogenetic trees resulting from Bayesian inference analyses were visualized and edited using FigTree v.1.4.3 [39].

In addition, twenty-nine *COI* gene sequences of the genus *Chironomus* from “GenBank” and “Barcode of Life Data Systems” (BOLD) were analyzed. Accession numbers of used sequences in GenBank and BOLD: *Ch. acutiventris* (Wülker, Ryser et Scholl, 1983) (AF192200.1), *Ch. annularius* (Meigen, 1818) (AF192189.1), *Ch. balatonicus* (Devai, Wulker et Scholl, 1983) (JN016826.1), *Ch. bernensis* (Wülker and Klötzli, 1973) (AF192188.1), *Ch. borokensis* (Kerkis, Filippova, Schobanov, Gunderina and Kiknadze, 1988) (AB740261), *Ch. cingulatus* (Meigen, 1830) (AF192191.1), *Ch. cingulatus* (Meigen, 1830) (AF192191.1), *Ch. commutatus* (Keyl, 1960) (AF192187.1), *C. dilutus* (Shobanov, Kiknadze and Butler, 1999) (KF278335.1), *Ch. entis* (Shobanov, 1989) (KM571024.1), *Ch. heteropilicornis* (Wuülker, 1996) (MK795770.1), *Ch. luridus* (Strenzke, 1959) (AF192203), *Ch. maturus* (Johannsen, 1908) (DQ648204.1), *Ch. melanescens* (Keyl, 1961) (MG145351), *Ch. novosibiricus* (Kiknadze, Siirin and Kerkis, 1993) (AF192197.1), *C. pallidivittatus* (Malloch, 1915) (AF110164), *Ch. piger* (Strenzke, 1959) (AF192202.1), *Ch. pilicornis* (Fabricius, 1787) (HM860166.1), *Ch. plumosus* (Linnaeus, 1758) (KF278217.1), *Ch. pseudothummi* (Strenzke, 1959) (KC250754), *Ch. riihimakiensis* (Wuülker, 1973) (MZ660756, MZ659595), *Ch. riparius* (Meigen, 1804) (KR756187.1), *Ch. sokolovae* (Istomina, Kiknadze and Siirin, 1999) (MW471100), *Ch. sororius* (Wülker, 1973) (MZ324811), *C. tentans* (Fabricius, 1805) (AF110157.1), *Ch. tenuistylus* (Brundin, 1949) (AF192201), *Ch. tuvanicus* (Kiknadze, Siirin et Wülker, 1993) (AF192196.1), *Ch. usenicus* (Loginova and Belyanina, 1994) (JN016806.1), *Ch. whitseli* (Sublette and Sublette, 1974) (KR683438.1). The *COI* sequence of *Drosophila melanogaster* (Meigen, 1830) (HQ551913) was used as an outgroup in phylogenetic analysis. 

## 3. Results

### 3.1. Morphological Characters of Ch. laetus from the Yamal Peninsula

The morphological characteristics of the fourth-instar larvae are presented in Figure 2. Most of the morphological characteristics correspond to the original description of *Ch. laetus* [4]. Body length is 10–14 mm. The head capsule is dark brownish in color. The frontoclypeus is dark with blurred borders. Occipital scleritis is dark. The antenna blade is extended to the middle of the fourth segment, but sometimes it may be higher than the fifth segment; the ring organ is located at a length of 2/3 of the basal segment length (Figure 2a). Ventromental plates with flat frontal edges (Figure 2b). The fourth and fifth teeth of the mentum are in the same line as the third tooth and a little lower (Figure 2e). The fourth tooth of the mandible is small and lighter colored (Figure 2d). Lateral tubuli at segment VII are absent, while two pairs of ventral tubuli at segment VIII are present, with rounded apeces (Figure 2c).

### 3.2. Karyotypes of Chironomus from the Yamal Peninsula

#### 3.2.1. Karyotype of *Ch*. sp. Le1

The karyotype (half karyotype) of *Ch.* sp. Le1 was studied only in single hybrid larvae (Figure 3). The chromosome set of the *Ch.* sp. Le1 is *n* = 4. The chromosome arm combination matches to the “thummi” cytocomplex: AB CD EF G. The chromosomes AB and CD are metacentric, EF is submetacentric, and G is telocentric. It was noted [17] that the arm G has the appearance of an amphora (Figure 3), which is typical for species of the *Ch. riihimakiensis* group. The bands in the centromeric region are heterochromatinized, and are not joined into the chromocenter [19,21]. The karyotype of the species is characterized by the presence of several nuclei and Balbiani rings, typical for the *Ch. riihimakiensis* group. The chromosome banding sequences of *Ch*. sp. Le1 are very similar to those of *Ch. riihimakiensis*, with the main differences being a larger (heterochromatinized) centromeric band and two dominant sequences, A2 and F2; it is thought that *Ch*. sp. Le1 is the northern variant of *Ch. riihimakiensis* [17,19,40].

We found only one genotypic combination, Sp. Le1A2.B1.C1.D1.E1.F2.G1., with seven known banding sequences that corresponded to *Ch.* sp. Le1 [17,19,21,40]:

**Arm A**. sp. Le1A2 1a-2c 10a-11e 9e-a 2d-3i 12c-a 8g-4a 13a-19f C;

**Arm B**. One unmapped banding sequence;

**Arm C.** sp. Le1C1 1a-6b 11c-8a 15e-11d 6gh 17a-16a 7d-a 6f-c 17b-22g C;

**Arm D.** sp. Le1D1 1a-3g 11a-18f 7g-4a 10e-8e 18g-24g C;

**Arm E**. sp. Le1E1 1a-3e 5a-10b 4h-3f 10c-13g C;

**Arm F**. sp. Le1F2 1a-8c 12a-17d 10d-8d 11i-a 18a-23f C;

**Arm G**. One unmapped banding sequence.

#### 3.2.2. Karyotype of *Ch. laetus*

The karyotype of *Ch. laetus* has not previously been mapped. It was noted, that *Ch. laetus* by chromosome arm combination correspond to “pseudothummi” cytocomplex—AE CD BF G [4], but a valid arm combination is AE BC DF G and this is an unknown cytocomplex, which we propose to named “laetus”. The chromosomes BC and DF are metacentric, AE is submetacentric, and G is telocentric (Figure 4). Visually we identified nucleoli (or puffs) in the arms D, F and G, which is typical for the *Ch. riihimakiensis* group [17]. 

In all studied larvae we found only one genotypic combination laeA1.1.E1.1.B1.1.C1.1.D1.1.F1.1.G1.1. We carried out preliminary mapping of chromosomal arms because all mapped banding sequences (or very similar) are already known for species *Ch. riihimakiensis* group: *Ch.* sp. Al1 (Kiknadze, Siirin and Wülker, 1992) and *Ch.* sp. Tu1 (Kiknadze, Siirin and Wülker, 1992) [20,41]:

**Arm A.** laeA1 = Al1 = LeA1 1a-2c 10a-12c 3i-2d 9e-4a 13a-19f C;

**Arm E.** laeE1 = Le1E1 1a-3e 5a-10b 4h-3f 10c-13g C;

**Arm B.** laeB1 = Al1B2 1a-2d 22d-21a 7a-8a 3c-2e 20g-18a 6f-a 4a-5e13g-8a 17d-14a 23-28a C;

**Arm C.** laeC1 = Al1C1 = sorC2 = abeC1 1a-6b 11c-8a 15e-11d 6gh 17a-16a 7d-a 6f-c 17b-22g C;

**Arm D.** laeD1 = Tu1D2 1a-3g 17f- 11a 18a-d 7g-4a 10e-8a 18e-24g C;

**Arm F.** laeF1 = sorF1 = abeF1 1a-10d 17d-11a 18a-23f C;

**Arm G.** One unmapped banding sequence.

#### 3.2.3. Karyotype of the Hybrid *Ch. laetus* × *Ch.* sp. Le1

In one preparation, we found the genotypic combination AE AB BC CD DF EF G. It was a result of the hybridization of two species from different cytocomplexes: AB CD EF G (“thummi”) and AE BC DF G (“laetus”). In the telomere region of arms E and F, we observe conjugation (Figure 5). The presence of the asynapsis indicates significant differences between the banding sequences of the parent species.

### 3.3. COI Gene Sequences and Phylogenetic Analysis of Chironomus from the Yamal Peninsula

We obtained four *COI* gene sequences for the studied larvae. Percentage of the nucleotides: A: 26%; T: 37%; G: 18%; and C: 19%. All sequences were deposited in GenBank with accession numbers: OP205478: *Ch. laetus* × *Ch.* sp. Le1 hybrid, length 658 bp. OP205321: *Ch. laetus* from an unnamed lake, length 694 bp. OP205477 and OP199059—*Ch. laetus* from Pidarmato Lake, lengths of 680 bp and 658 bp, respectively. 

#### 3.3.1. Genetic Distances Obtained with K2P

The pairwise genetic distances between the obtained sequences were calculated by the K2P model [35]. The distance between the hybrid of *Ch. laetus* × *Ch.* sp. Le1 and *Ch. laetus* sequences was 0.2%, and no differences were found between sequences of *Ch. laetus* from Pidarmato Lake and Unnamed Lake (Table 1).

As previously shown, the studied species are members of the *Ch. riihimakiensis* group, and two *COI* gene sequences from *Ch. riihimakiensis*, MZ660756 and MZ659595, were found in GenBank; they were sampled in Finland, at the same locality (60°23′52.8″ N and 23°05′56.4″ E). The genetic distance between the sequences of *Ch. riihimakiensis* (MZ660756) and *Ch. laetus* was 2.5%, and the *Ch. laetus* × *Ch.* sp. Le1 hybrid was 2.3%. The genetic distance between sequences of *Ch. riihimakiensis* MZ660756 and MZ659595 was 12.4%, which is more than the accepted threshold of 3% [12,13,14,15,16]. The interesting point is that the distance between the sequences of *Ch. laetus* and species from the other members of the *Ch. riihimakiensis* group, *Ch. novosibiricus* and *Ch. tuvanicus*, is about 14%. One of the sequences of *Ch. riihimakiensis* (MZ659595) has a high degree of similarity with the sequence of *Ch. tenuistylus* (AF192201), approximately 3.1%.

#### 3.3.2. The Analysis of the Phylogenetic Tree

The phylogenetic tree obtained by Bayesian inference showed groups of related species of *Chironomus* (Figure 6). Here the situation is repeated (see Table 1). All sequences of *Ch. laetus*, including the hybrid, are combined into one cluster with *Ch. riihimakiensis* (MZ660756). Another sequence of *Ch. riihimakiensis* (MZ659595) forms a distinct cluster with the sequence of *Ch. tenuistylus* (AF192201). The two species from the true *Ch. riihimakiensis* group, *Ch. novosibiricus* and *Ch. tuvanicus*, form another distinct (independent) cluster.

## 4. Discussion

During the rare investigation of rivers and lakes located in the Arctic zone, every time it is possible to make interesting findings about chironomids (a new species, banding sequences, etc.) [4,15,21,42]. 

Chironomids are one of the essential components in most bodies of water. Because of this, they are convenient objects for environmental monitoring and the study of biodiversity. The main obstacle to this is the difficulty of sibling-species identification using only larval [18,19], pupal, and imaginal morphology. In order to avoid misidentification, it is suggested to use a comprehensive approach including cytogenetic and molecular-genetic analysis [8,12,13]. 

In Pidarmato Lake and an unnamed lake in the Yamal Peninsula, we collected larvae of *Chironomus* with a dark brown head capsule. They possess the morphological characteristics that are typical for the *Chironomus riihimakiensis* group, and at first, we were not sure of the species identification accuracy.

Cytogenetic analysis of one larva revealed a very interesting karyotype (chromosome set), which consisted of two different species’ chromosomes. It was a hybrid of *Ch.* sp. Le1, a northern variant of *Ch. riihimakiensis* [17,40], and another species from the unknown cytocomplex, so we thought that the species was also new. In one paper devoted to the study of three sympatric species of *Chironomus*, we found the figure with the very similar karyotype of *Ch. laetus* [4]. 

We did not find *Chironomus* sp. Le1 among larvae from the studied lakes; only one hybrid was found with *Ch. laetus*. In the previously studied populations, the genotypic combination Le1A2.B1.C1.D1.E1.F2.G1. is also dominant [17]. Our findings confirm the circumpolar distribution of *Ch.* sp. Le1 [17,19]. 

The karyotype of *Ch. laetus* from the Yamal Peninsula is very similar to the first photomap presented by Belyanina and Filinkova on Figure 6a in [4]. However, the authors indicated the wrong “pseudothummi” cytocomplex with the chromosome arm combination AE CD BF G [4]. They have mixed up the arms of C and F in CD and BF combinations. It would be correct to specify combinations of BC and DF. Visually, we noted differences only in the absence of nucleoli in arm E of our specimens. This may be an effect of the environmental conditions. In *Chironomus* species evolution, of great significance is the reciprocal translocation of whole chromosomal arms and, as a result, the formation of diverse arm combinations. Based on this, all *Chironomus* species are grouped into 17 different cytocomplexes [19,43]. We propose the name “laetus” for the new cytocomplex because *Ch. laetus* is the first species with the arm combinations AE BC DF G. 

We performed a preliminary analysis of the banding sequences of *Ch. laetus* and found their similarity with the already known sequences of the primitive *Chironomus riihimakiensis* group. The most conservative banding sequences in *Chironomus* are located in arms E and F [31,44], and this species is no exception. The laeE1 banding sequence is very similar to most species from the cytocomplexes «thummi» and «pseudothummi», including Le1E1, Al1E1, and pluE1. The laeF1 banding sequence is very similar to sorF1, abeF1, and borF1; note that the part of the sequence “17d-11a 18a-23f” excluding the telomere region, is characteristic of most of the species *Chironomus* [43]. The banding sequence laeA1 is very similar to Le1A1, Al1A1, and a few species from the cytocomplexes “thummi” and “pseudothummi”. The sequences in arm B are more species-specific [45]. With reference to laeB1, only with the Al1B2 sequence we did find a high degree of similarity. The banding sequence laeC1 is very similar to Al1C1, pilC1, sorC2, and abeC1. The banding sequence laeD1 is similar to Tu1D2.

The occurrence of interspecific hybridization in chironomids is a quite rare event [23,24,25,27,28]. Our finding of a hybrid between *Chironomus* from different cytocomplexes is presented for the first time. This means that there is no complete reproductive isolation between two closely related species. The new cytocomplex (and the species) are probably very young, as *Ch. laetus* is not only close to other species in the *Ch. riihimakiensis* group but can also form hybrids. This is confirmed by the conjugation of homologues in the telomere regions of the two most conservative arms, E and F. The hybrid species’ chromosome sizes are very similar; the morphological characteristics of centromeric bands, nucleoli, and Balbiani rings position correspond to species-specific features (Figure 5). 

We have no opportunity to analyze the *COI* gene sequence of *Ch.* sp. Le1, because no larvae of this species were found to be hybrid. The hybrid (OP205478) and *Ch. laetus* sequences (OP205477, OP199059, and OP205321) differ by 0.2% or one nucleotide. We can assume that the *COI* gene sequence was from *Ch.* Sp. Le1, the maternal organism. Otherwise, it might be the result of nucleotide polymorphism; a similar case was noted in *Ch. bonus* when each of the three sequences had a different haplotype [16].

During the comparison of the genetic distance between the sequences of *Ch. laetus* and *Ch. riihimakiensis* (MZ660756 and MZ659595) from Finland, we received two different values, which are 2.5 and 13.4%. The distance between the two sequences of *Ch. riihimakiensis* was 12.4%, which is much more than the 3% accepted interspecific threshold [12,13]. It could be a result of the misidentification by morphological characteristics of one of them or both [8,9]. We can assume that they are two different species. We have not found any sequences of the species from the *Ch. riihimakiensis* group, and existing data is not enough to make final conclusions about the status of the species on the evolutionary tree. Nevertheless, analysis of one short segment of the *COI* gene is not enough to make final inferences [8]. The high degree of similarity between *Ch. laetus* banding sequences, larval morphological features, and the event of hybridization with the *Ch. riihimakiensis* group suggests a close relationship. At first, grouping was subjective (artificial) and had no precise criteria, so the species were united by some similarity of morphology and karyotype [43]. The position of *Ch. laetus* on the Bayesian tree in separate clusters from *Ch. riihimakiensis* group suggests the presence of the molecular-genetic subgroup of the species. 

We cannot ignore the presence of 48 *COI* gene sequences of *Chironomus* sp. with a 99–100% match with *Ch. laetus* in the GenBank, for example, *Ch*. sp. MN676646.1 (Canada, Cambridge Bay, 69°13′09.8″ N, 104°55′12.0″ W). This is a great opportunity to perform a preliminary investigation of the Arctic zone’s biodiversity with subsequent extension to morphological and cytogenetic examination techniques.

## Figures and Tables

**Figure 1 insects-13-01112-f001:**
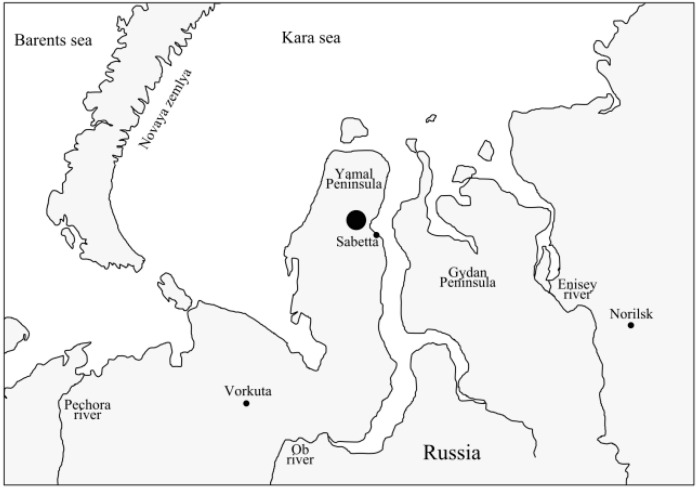
Locality of *Ch. laetus* and *Ch. laetus* × *Ch.* sp. Le1 in the Yamal Peninsula, Russia. The collection site is marked by a big black circle.

**Figure 2 insects-13-01112-f002:**
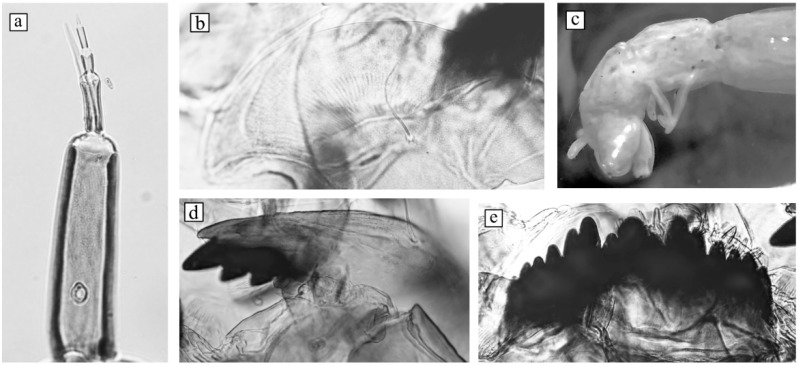
Larval morphology of *Ch. laetus* from the Yamal Peninsula, Russia. **a**—antenna, **b**—ventromental plate, **c**—abdominal apex, **d**—mandible, and **e**—mentum.

**Figure 3 insects-13-01112-f003:**
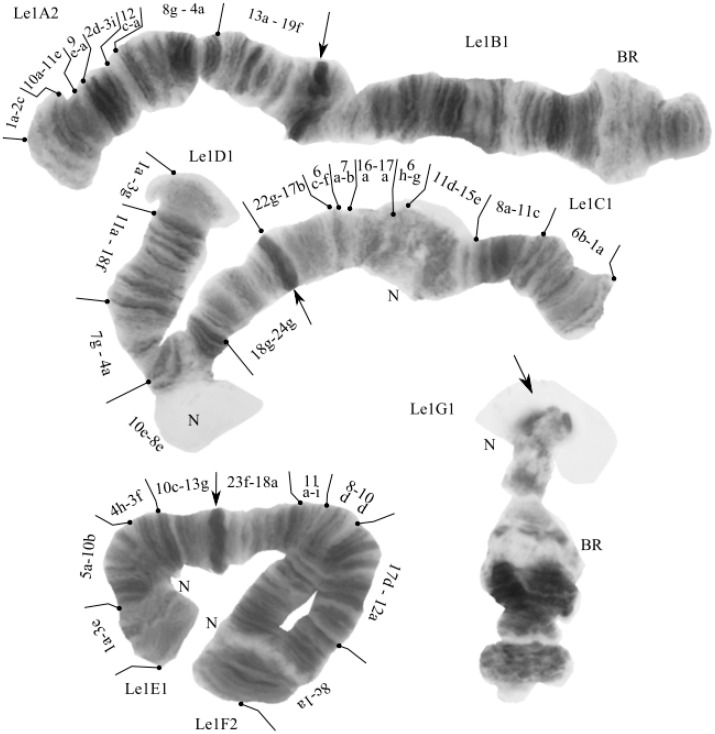
Part of the *Chironomus* sp. Le1 karyotype from a hybrid of *Ch.* sp. Le1 × *Ch. laetus* from the Yamal Peninsula, Russia. Arrows indicate a centromeric bands; Le1A2, Le1B1, etc., genotypic combinations of banding sequences in chromosome arms; BR—Balbiani rings; N—nucleous.

**Figure 4 insects-13-01112-f004:**
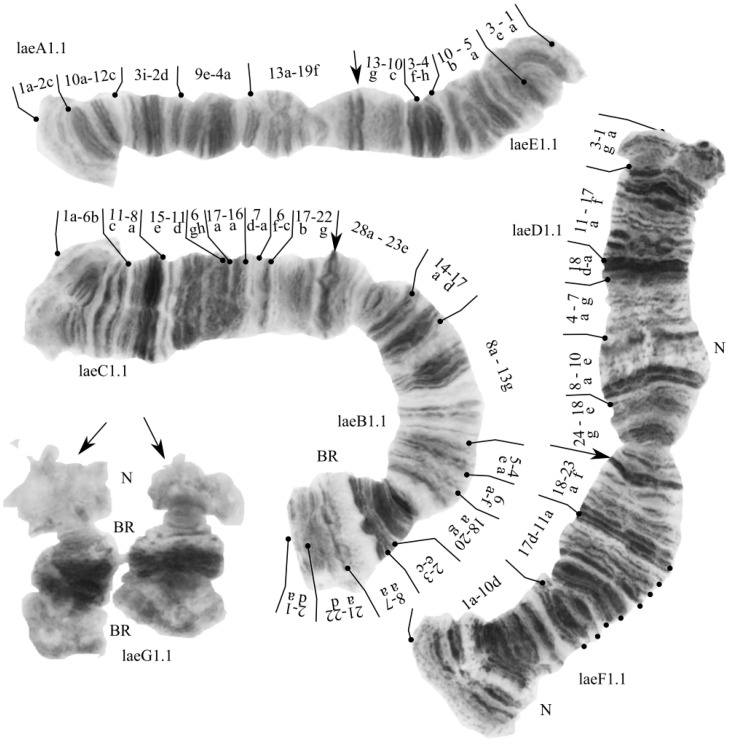
Karyotype of *Chironomus laetus* from the Yamal Peninsula, Russia. Arrows indicate centromeric bands; laeA1, laeE1, etc., genotypic combinations of banding sequences in chromosome arms; BR—Balbiani rings; N—nucleous.

**Figure 5 insects-13-01112-f005:**
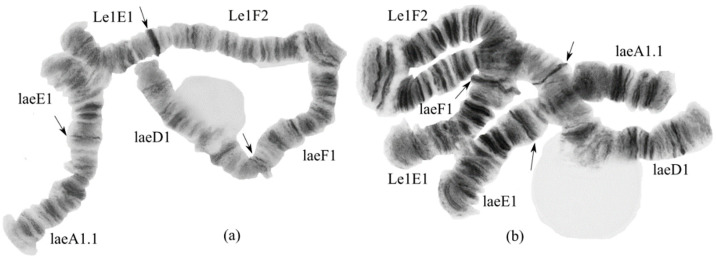
Chromosome arms of a *Ch.* sp. Le1 × *Ch. laetus* hybrid from the Yamal Peninsula, Russia. Arrows indicate centromeric bands; laeA1, Le1E1, etc., genotypic combinations of banding sequences in chromosome arms. (**a**,**b**): Chromosome arms of a *Ch.* sp. Le1 × *Ch. laetus* hybrid from the Yamal Peninsula, Russia.

**Figure 6 insects-13-01112-f006:**
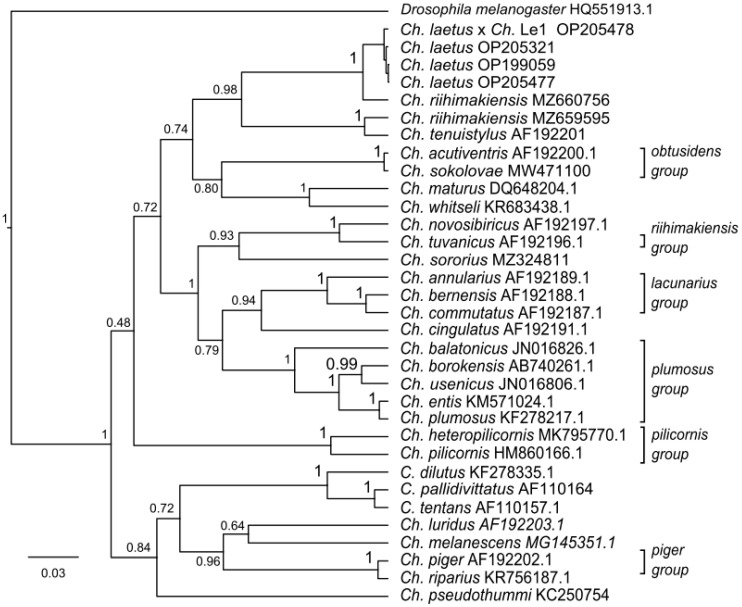
Bayesian tree of the analyzed samples of *Chironomus* spp. inferred from *COI* sequences. The species name, GenBank accession number, and group name are shown to the right of the branches. Support values are given if they exceed 0.3. The numbers at the nodes indicate posterior probabilities.

**Table 1 insects-13-01112-t001:** The pairwise genetic distances (Kimura-2p, %) between *COI* gene sequences of Chironomus.

	*Ch. laetus × Ch.* sp. Le1OP205321	*Ch. laetus* OP205321	*Ch. laetus* OP205477	*Ch. laetus* OP199059	*Ch. riihimakiensis* MZ660756	*Ch. riihimakiensis* MZ659595	*Ch. tenuistylus* AF192201	*Ch. novosibiricus* AF192197
*Ch. laetus* OP205321	0.2							
*Ch. laetus* OP205477	0.2	0.0						
*Ch. laetus* OP199059	0.2	0.0	0.0					
*Ch. riihimakiensis* MZ660756	2.3	2.5	2.5	2.5				
*Ch. riihimakiensis* MZ659595	13.1	13.4	13.4	13.4	12.4			
*Ch. tenuistylus* AF192201	12.8	13.1	13.1	13.1	12.2	3.1		
*Ch. novosibiricus* AF192197	14.9	14.6	14.6	14.6	14.8	17.7	17.2	
*Ch. tuvanicus* AF192196	13.9	14.1	14.1	14.1	13.4	17.0	16.5	5.4

## Data Availability

The data supporting the reported results can be obtained upon request from the corresponding author (V.V.B.).

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
