# Peer review of "Karyotypes and COI Gene Sequences of Chironomus sp. Le1 (Kiknadze and Salova, 1996), Ch. laetus (Belyanina and Filinkova, 1996) and Their Hybrid from the Yamal Peninsula, Arctic Zone of Russia"

_insects, 2022, doi:10.3390/insects13121112_

Round 1

Reviewer 1 Report

1. Page 2, line 54: I suppose that the word in brackets has to be replaced with “difficult”.

2.  Page 2, line 58: I suppose that the line has to be rephrased this way: Recently, by this method, the biodiversity of invertebrates has been studied in Canada [5], Norway [6], and South Korea [7].

3. Page 2, line 76: I suppose that the line has to be rephrased this way: After precise analysis, we understand that this is a hybrid.

4.  Page 3, line 104: Why don’t you refer to the most recent publication on the topic, like Kiknadze et al. (2016), where the Chironomus riihimakiensis group includes just the following species:

Chironomus riihimakiensis Wülker, 1973

Chironomus sp. Al1 Kiknadze, Siirin & Wülker, 1992

Chironomus sp. Le1 Kiknadze & Salova, 1996

Chironomus tuvanicus Kiknadze, Siirin & Wülker, 1992

Chironomus sp. Tu1 Kiknadze, Siirin & Wülker, 1993?

Why do you use slightly outdated references like Kiknadze et al. 1992 and Shobanov 2000?

5.  Page 3, lines 106-107: If it was you who calculated the distance of 12% between specimens of Ch. riihimakiensis (MZ659595 and MZ660756), this line has to be in the Results part of the manuscript. Otherwise, if you got this data from some publication, you have to provide the reference.

6.  Fig 3. In arm A, the designation of one region needs to be corrected. In the photo designation of 8q-4a region, it needs to be replaced by 8g-4a.

7.  Page 8, lines 259-260: “Percentage of the nucleotides A: 26; T: 37; G: 18; C: 19”. What do you mean with this line? Is this to say that the four COI gene sequences all had the same haplotype with the same percentage of nucleotides? The NCBI sequences clearly show that the three Ch. laetus sequences have one haplotype and the hybrid has another. The difference is very small (one nucleotide), but it is present. It would be better to mention these details in the text of the manuscript and provide the position number of this mutation.

8. Figure 6. You designated the node in the Bayesian tree where Ch. novosibiricus and Ch. tuvanicus are located as the Ch. riihimakiensis group. At the same time, according to Kiknadze et al. 1992, Shobanov 2000, and Kiknadze et al. (2016), the species Ch. novosibiricus does not belong to the group. Did you do this because of the K2P distances in Table 1, where Ch. novosibiricus is closest to Ch. tuvanicus (5.4%)? Is it your assumption? If so, then you should mention this in the text of the manuscript and explain your considerations.

9. Figure 6. For a better understanding of the tree topology, it would be good to add information from which countries the sequences were provided, especially the sequences of Ch. riihimakiensis, Ch. laetus, and Ch. laetus x Ch. sp. Le1 hybrid.

Author Response

We would like to express our appreciation for the Reviewers comments to our manuscript!

Point-by-point response.

The changes are highlighted in yellow.

  1. Page 2, line 54: I suppose that the word in brackets has to be replaced with “difficult”.

- corrected

  1. Page 2, line 58: I suppose that the line has to be rephrased this way: Recently, by this method, the biodiversity of invertebrates has been studied in Canada [5], Norway [6], and South Korea [7].

- corrected

  1. Page 2, line 76: I suppose that the line has to be rephrased this way: After precise analysis, we understand that this is a hybrid.

- corrected

  1. Page 3, line 104: Why don’t you refer to the most recent publication on the topic, like Kiknadze et al. (2016), where the Chironomus riihimakiensis group includes just the following species: Chironomus riihimakiensis Wülker, 1973 Chironomus sp. Al1 Kiknadze, Siirin & Wülker, 1992 Chironomus sp. Le1 Kiknadze & Salova, 1996 Chironomus tuvanicus Kiknadze, Siirin & Wülker, 1992 Chironomus sp. Tu1 Kiknadze, Siirin & Wülker, 1993? Why do you use slightly outdated references like Kiknadze et al. 1992 and Shobanov 2000?

- corrected. Ch. pilicornis and Ch. heteropilicornis – deleted.

  1. Page 3, lines 106-107: If it was you who calculated the distance of 12% between specimens of Ch. riihimakiensis (MZ659595 and MZ660756), this line has to be in the Results part of the manuscript. Otherwise, if you got this data from some publication, you have to provide the reference.

- In the “Results”. Thank you!

  1. Fig 3. In arm A, the designation of one region needs to be corrected. In the photo designation of 8q-4a region, it needs to be replaced by 8g-4a.

- corrected

  1. Page 8, lines 259-260: “Percentage of the nucleotides A: 26; T: 37; G: 18; C: 19”. What do you mean with this line? Is this to say that the four COI gene sequences all had the same haplotype with the same percentage of nucleotides? The NCBI sequences clearly show that the three Ch. laetus sequences have one haplotype and the hybrid has another. The difference is very small (one nucleotide), but it is present. It would be better to mention these details in the text of the manuscript and provide the position number of this mutation.

- The percentage of nucleotides is convenient for further comparison at the level of genera.

“… provide the position number of this mutation” - This is not the purpose of this investigation and requires further examination.

  1. Figure 6. You designated the node in the Bayesian tree where Ch. novosibiricus and Ch. tuvanicus are located as the Ch. riihimakiensis group. At the same time, according to Kiknadze et al. 1992, Shobanov 2000, and Kiknadze et al. (2016), the species Ch. novosibiricus does not belong to the group. Did you do this because of the K2P distances in Table 1, where Ch. novosibiricus is closest to Ch. tuvanicus (5.4%)? Is it your assumption? If so, then you should mention this in the text of the manuscript and explain your considerations.

- Figure 6. - corrected.

  1. Figure 6. For a better understanding of the tree topology, it would be good to add information from which countries the sequences were provided, especially the sequences of Ch. riihimakiensis, Ch. laetus, and Ch. laetus x Ch. sp. Le1 hybrid.

- I think this is unnecessary information on the graphic; there is only one country - Finland.

- line 271. In the text “…two COI gene sequences from Ch. riihimakiensis, MZ660756 and MZ659595, were found in GenBank; they were sampled in Finland, at the same locality (60°23'52.8"N 23°05'56.4"E)”.

Reviewer 2 Report

The article describes the morphological, chromosomal and molecular characteristics of six Chironomus larvae. The sample size is too small, which compromises the robustness of the data. In addition, the presentation of the article contains a number of imprecise sentences, which may be, in part, due to problems in the English translation. This makes it difficult to understand the relevance of a few original data that the article contains. The figures have low quality and some oversights such as lack of scale, absence of geographic coordinates etc. Perhaps it would be better to present it as a short  communication, after a deep restructuring.

Author Response

We would like to express our appreciation for the Reviewers comments to our manuscript!

Thank you for your time!

The changes are highlighted in yellow.

The article describes the morphological, chromosomal and molecular characteristics of six Chironomus larvae. The sample size is too small, which compromises the robustness of the data.

- In similar papers, especially with materials from hard-to-reach places, even one sample is interesting. Several samples are more typical for specialized hydrobiological studies. Here we obtained a lot of information out of six samples.

In addition, the presentation of the article contains a number of imprecise sentences, which may be, in part, due to problems in the English translation. This makes it difficult to understand the relevance of a few original data that the article contains.

- Yes. Thank you, we will correct it.

The figures have low quality and some oversights such as lack of scale, absence of geographic coordinates etc.

- The figures are of the usual quality for the article, all the details are clearly visible. A scale is not required. The geographical coordinates are in the text.

Perhaps it would be better to present it as a short communication, after a deep restructuring.

- There are a lot of figures and tables for a short communication. In addition, I tried to shorten the text, but without a description of the previous studies, in my opinion, it is impossible. We present much information for the first time.

Reviewer 3 Report

Dear Authors,

This paper has merit for publication, but you will need to address many grammatical errors throughout the text. I think this will take some time and suggest that you seek the assistance of a colleague, perhaps someone who studies chironomids and who will help in exchange for mention in the acknowledgments. 

The paper introduction also needs to be revised. In the current draft, your objective and/or aim and goals are not clear. Is this a paper addressing new species or species divisions and phylogenetic placement or is it a paper exploring the biodiversity of arctic or far northern lakes using molecular tools. I think that it is the second. If so, then I suggest the following order for the revision of the introduction:

Unique ecosystems of the study lakes

Numerical dominance, richness, and functional importance of chironomids in these types of lakes

Need to better understand biodiversity in these lakes to better understand function

Problems describing biodiversity in Chironomus

Use of molecular tools to resolve these problems

Objective is to define biodiversity of common species of Chironomus in these study lakes

Goals to apply molecular tools to determine species diversity/resolve species overlap, etc.

I hope this helps and look forward to seeing a revision of the paper.

Author Response

We would like to express our appreciation for the Reviewers comments to our manuscript!

We agree with you that grammatical errors need to be corrected.

I'm not sure I understand your comments about the structure of the article. It seems to me that it basically matches your structure. If I add more information about lakes and their biodiversity, the article will be overloaded. I have used several of your comments for our article and will use them in the preparation of further studies.

Thank you very much!

The changes are highlighted in yellow.

Dear Authors,

This paper has merit for publication, but you will need to address many grammatical errors throughout the text. I think this will take some time and suggest that you seek the assistance of a colleague, perhaps someone who studies chironomids and who will help in exchange for mention in the acknowledgments. The paper introduction also needs to be revised. In the current draft, your objective and/or aim and goals are not clear. Is this a paper addressing new species or species divisions and phylogenetic placement or is it a paper exploring the biodiversity of arctic or far northern lakes using molecular tools. I think that it is the second. If so, then I suggest the following order for the revision of the introduction:

Unique ecosystems of the study lakes

Numerical dominance, richness, and functional importance of chironomids in these types of lakes

Need to better understand biodiversity in these lakes to better understand function

Problems describing biodiversity in Chironomus

Use of molecular tools to resolve these problems

Objective is to define biodiversity of common species of Chironomus in these study lakes

Goals to apply molecular tools to determine species diversity/resolve species overlap, etc.

I hope this helps and look forward to seeing a revision of the paper.

Reviewer 4 Report

First of all I should mention that the English must be corrected, better checked by native English speaker.

I also have some suggestion for improvement of the text and also some questions and sugestions for discussing of the material presented:

Simple Summary:

1)      Row 16: I suggest to change the sentences to: Due to the presence of giant polytene chromosomes in their salivary gland, it is more convenient to use cytogenetics for species identification.

2)      Row 20: chironomids (don’t use upper case later as it is not a latin name of the genus)

3)      Row 22: By cytogenetics, we identified the species as Ch. laetus, the first species with arm combinations AE BC DF G, and propose the name of a new cytocomplex - cytocomplex «laetus».

In this case I would like to specifically point out that the name of the cytocomplex should be just “laetus”, not Chiromomus “laetus”.

4)      Row 24: I suggest to change the sentences to: We also found a hybrid Ch. laetus x Ch. sp. 23 Le1, it is the first hybrid between species from different cytocomplexes.

5)      Row 26: The meaning of this sentence is not clear. Authors should add clarification about what subgroup they suspect there is in the species-group.

Abstract:

6)      Row 33: it is the first species…

7)      Row 33: We propose the name of a new cytocomplex – cytocomplex «laetus».

Introduction:

8)      Row 47: I suggest to change the sentence to this variant as in the original one can assume that all mentioned species were described only by morphological features.

Only a few species have been previously recorded from the region of investigation: Chironomus beljaninae Wülker, 1991 [1], Ch. plumosus (L., 1758), Ch. nigrifrons (Linevich et Erbaeva 1971) [2], and also a lot of unidentified Chironomus spp., which were described only on the basis of larval morphology [3].

9)      Row 50: It is not clear if all three species were described without karyotypes or just one?

10)  Row 54: I suggest to change the sentence: The main problem in the studies of Chironomus is the incorrect identification of species by morphological characteristics, especially at the larvae stage when reliable identification of some sibling species can be completely impossible.

11)  Row 57: should be “… the Barcoding, which uses…”

12)  Row 58: I suggest to change the sentence: Recently this method was used to study biodiversity of invertebrates in Canada [5], Norway [6] and South Korea [7].

13)  Row 71: I suggest to use "karyological analysis" instead of "cytogenetics"

14)  Row 76: I suggest to change the sentence to: After thorough analysis we came to the conclusion that it was an interspecies hybrid.

15)  Row 83: I suggest to change the sentences to: Ch. sp. Le1 was recorded from the delta of Lena River [19], Novaya Zemlya Islands, Dikson (Taimyr Peninsula), Alaska and Ellesmere in North America [17], and the Wrangel Island [22], which suggest that the species has circumpolar distribution [22,23].

16)  Row 88: I suggest to change the sentences to: But analysis of karyotype published in that work shoed that authors incorrectly identified chromosomal arms and the true combination was AE BC DF G, unknown (new) cytocomplex.

17)  Row 91: I suggest to change the sentences to: Only few cases of natural hybrids were recorded:

18)  Row 95: I suggest to change the sentences to: Several experimental hybrids were also obtained under laboratory conditions:

19)  Row 99: I suggest to change the sentences to: Hybrids between species that belong to the different cytocomplexes have not been reported previously.

20)  Row 101: I suggest to change the sentences to:  In this work we use complex approach to the species identification of Chironomus.

21)  Row 102: …of about…

22)  Row 109: Chironomids are…

23)  Row 110: Molecular-genetic approaches are very popular at present and they require high precision in identification of species in different levels (actually it is not clear what authors mean by “different levels”. Stages of development? Or different methods to identify? It the latter, I’d suggest to say “using different methods” instead of “in different levels”).

Materials and methods

24)  Row 137: (Folmer et al. 1994). should be changed to number as the rest of literuture sited? I also hasn’t found it in the sited literature.

Results

25)  Row 179: “correspond” probably would be a better term

26)  Row 180: Body length is…

27)  Row 182: but sometimes it may be…

28)  Row 185: the same line…

29)  Row 194: Authors state that they studied karyotype of Ch. sp. Le1 only in the hybrid larvae, yet they refer to Figure 3 where karyotype of normal larvae belonging to Ch. sp. Le1 is shown (with full set of chromosomes, 2n=8). Authors should clarify this point (I assume they show karyotype from another specimen found elsewhere for the easier understanding of karyotype features).

30)  Row 196: should be just "thummi" cytocomplex, without Chironomus

31)  Row 225: Why arm C is mentioned specifically? In the original description nucleoli said to be present in arms G and E only. If you refer to other species of the Ch. riihimakiensis group, you should clarify this, otherwise I suggest just to remove the mention of arm C.

32)  Row 226: Actually it is incorrect to state that “The main differences between karyotypes of Ch. laetus from the Yamal Peninsula population and the original description are the absence of the nucleoli in arm E and the presence of it in arm B” because the description by Belyanina and Filinkova was wrong in the first place. Moreover, it is clear from the katyotype they published that there were some big puffs (probably nucleoli) in arms D and F (incorrectly designated as C), so their karyotype is virtually identical to karyotype that is described by authors here. Also the sentence state that nucleoli was found in arm B, while just above it was D, F and G, so I ssume there is a mistake here. I’d suggest to authors to stress out more clearly that there were mistakes in the original description of karyotype published by Belyanina and Filinkove and discuss this in more detail.

Also I would like to see the information on what ground authors decided what puffs are nucleoli and what are not. Recently the localization of nucleoli for some species has been confirmed by molecular biological methods (in situ hybridization of rDNA), which is more precise that any other used previously, so if authors done the localization just visually it should be stated in the paper (as it means there could be more of nucleoli in karyotype and also some that are assumed to be nucleoli (like in arm F) might actually be just puffs).

33)  Row 230: … mapping of chromosomal arms…

34)  Row 233: what morphological characteristics? This should be clarified.

35)  Row 251: better to say 'we observe" instead of "we note"

36)  Row 281: in this case I would add that the second sequence of Ch. riihimakiensis most probably doesn't belong to the species and it was mistake in species identification.

37)  Row 290, figure 6: there is probably a mistake in naming as in pilicornis group there is Ch. heterodentatus. I assume it was supposed to be Ch. heteropilicornis? Please check this.

38)  Row 301: I would add here that identification of pupae and imago are also not 100% reliable as the problem of presence in GENEBANK of a lot of a lot of misidentified sequences appeared because imago was used for sequencing.

39)  Row 313; remove "Chironomus"

40)  Row 315: I suggest to change it to: Between larvae from studied lakes we did not find Chironomus sp. Le1, only one hybrid with Ch. laetus.

41)  Row 319: As I mentioned above, the discussion of this should be changed completley as katyotypes are actually identical, it was description of it that was wrong.

42)  Row 326: remove Chironomus

43)  Row 334: excluding

44)  Row 347: Here I would also add a suggestion that the new cytocomplex (and the species) probably is very young as Ch. laetus not only is close to other species from XH. riihimakiensis group but can form hybrids. Actually the finding of this hybrid is very unique as we indid never saw such hybridization between species from different cytocomplexes (actually, mostly only sibling-species or very closely related species can form hybrids). So this fingding is a gem of this work.

45)  Raw 349: no larvae of this species was found

46)  Row 365, 367, 369, 373: name of the siblinb-species group should be written as "Ch. riihimakiensis", without ''''

47)  Row 377: suggests the presence

Author Response

We would like to express our appreciation for the Reviewers comments to our manuscript!

The changes are highlighted in yellow.

First of all I should mention that the English must be corrected, better checked by native English speaker.

- Corrected.

I also have some suggestion for improvement of the text and also some questions and sugestions for discussing of the material presented:

Simple Summary:

1)      Row 16: I suggest to change the sentences to: Due to the presence of giant polytene chromosomes in their salivary gland, it is more convenient to use cytogenetics for species identification.

- Corrected.

2)      Row 20: chironomids (don’t use upper case later as it is not a latin name of the genus)

- Corrected.

3)      Row 22: By cytogenetics, we identified the species as Ch. laetus, the first species with arm combinations AE BC DF G, and propose the name of a new cytocomplex - cytocomplex «laetus».

In this case I would like to specifically point out that the name of the cytocomplex should be just “laetus”, not Chiromomus “laetus”.

- Corrected.

4)      Row 24: I suggest to change the sentences to: We also found a hybrid Ch. laetus x Ch. sp. 23 Le1, it is the first hybrid between species from different cytocomplexes.

- Corrected.

5)      Row 26: The meaning of this sentence is not clear. Authors should add clarification about what subgroup they suspect there is in the species-group.

- Corrected. “Molecular genetic data suggests the presence of a subgroup with Ch. laetus”.

Abstract:

6)      Row 33: it is the first species…

- Corrected.

7)      Row 33: We propose the name of a new cytocomplex – cytocomplex «laetus».

- Corrected.

Introduction:

8)      Row 47: I suggest to change the sentence to this variant as in the original one can assume that all mentioned species were described only by morphological features.

Only a few species have been previously recorded from the region of investigation: Chironomus beljaninae Wülker, 1991 [1], Ch. plumosus (L., 1758), Ch. nigrifrons (Linevich et Erbaeva 1971) [2], and also a lot of unidentified Chironomus spp., which were described only on the basis of larval morphology [3].

- Corrected.

9)      Row 50: It is not clear if all three species were described without karyotypes or just one?

- Corrected. “… of their karyotypes …”

10)  Row 54: I suggest to change the sentence: The main problem in the studies of Chironomus is the incorrect identification of species by morphological characteristics, especially at the larvae stage when reliable identification of some sibling species can be completely impossible.

- Corrected.

11)  Row 57: should be “… the Barcoding, which uses…”

- Corrected.

12)  Row 58: I suggest to change the sentence: Recently this method was used to study biodiversity of invertebrates in Canada [5], Norway [6] and South Korea [7].

- Corrected.

13)  Row 71: I suggest to use "karyological analysis" instead of "cytogenetics"

- Corrected.

14)  Row 76: I suggest to change the sentence to: After thorough analysis we came to the conclusion that it was an interspecies hybrid.

- Corrected.

15)  Row 83: I suggest to change the sentences to: Ch. sp. Le1 was recorded from the delta of Lena River [19], Novaya Zemlya Islands, Dikson (Taimyr Peninsula), Alaska and Ellesmere in North America [17], and the Wrangel Island [22], which suggest that the species has circumpolar distribution [22,23].

- Corrected.

16)  Row 88: I suggest changing the sentences to: But analysis of karyotype published in that work shoed that authors incorrectly identified chromosomal arms and the true combination was AE BC DF G, unknown (new) cytocomplex.

- Corrected.

17)  Row 91: I suggest changing the sentences to: Only few cases of natural hybrids were recorded:

- Corrected.

18)  Row 95: I suggest to change the sentences to: Several experimental hybrids were also obtained under laboratory conditions:

- Corrected.

19)  Row 99: I suggest to change the sentences to: Hybrids between species that belong to the different cytocomplexes have not been reported previously.

- Corrected.

20)  Row 101: I suggest to change the sentences to:  In this work we use complex approach to the species identification of Chironomus.

- Corrected.

21)  Row 102: …of about…

- Corrected.

22)  Row 109: Chironomids are…

- Corrected.

23)  Row 110: Molecular-genetic approaches are very popular at present and they require high precision in identification of species in different levels (actually it is not clear what authors mean by “different levels”. Stages of development? Or different methods to identify? It the latter, I’d suggest to say “using different methods” instead of “in different levels”).

- Molecular-genetic approaches are very popular last time and this requires high precision in identifying the species in different stages of development.

Materials and methods

24)  Row 137: (Folmer et al. 1994). should be changed to number as the rest of literuture sited? I also hasn’t found it in the sited literature.

- Corrected.

Results

25)  Row 179: “correspond” probably would be a better term

- Corrected.

26)  Row 180: Body length is…

- Corrected.

27)  Row 182: but sometimes it may be…

- Corrected.

28)  Row 185: the same line…

- Corrected.

29)  Row 194: Authors state that they studied karyotype of Ch. sp. Le1 only in the hybrid larvae, yet they refer to Figure 3 where karyotype of normal larvae belonging to Ch. sp. Le1 is shown (with full set of chromosomes, 2n=8). Authors should clarify this point (I assume they show karyotype from another specimen found elsewhere for the easier understanding of karyotype features).

- No, this is the single specimen. I do not know why the Le1F2 has such a structure.

“Figure 3. Part of the hybrid karyotype belonging to Chironomus sp. Le1 …”.

30)  Row 196: should be just "thummi" cytocomplex, without Chironomus

- Corrected.

31)  Row 225: Why arm C is mentioned specifically? In the original description nucleoli said to be present in arms G and E only. If you refer to other species of the Ch. riihimakiensis group, you should clarify this, otherwise I suggest just to remove the mention of arm C.

- removed.

32)  Row 226: Actually it is incorrect to state that “The main differences between karyotypes of Ch. laetus from the Yamal Peninsula population and the original description are the absence of the nucleoli in arm E and the presence of it in arm B” because the description by Belyanina and Filinkova was wrong in the first place. Moreover, it is clear from the katyotype they published that there were some big puffs (probably nucleoli) in arms D and F (incorrectly designated as C), so their karyotype is virtually identical to karyotype that is described by authors here. Also the sentence state that nucleoli was found in arm B, while just above it was D, F and G, so I ssume there is a mistake here. I’d suggest to authors to stress out more clearly that there were mistakes in the original description of karyotype published by Belyanina and Filinkove and discuss this in more detail.

Also I would like to see the information on what ground authors decided what puffs are nucleoli and what are not. Recently the localization of nucleoli for some species has been confirmed by molecular biological methods (in situ hybridization of rDNA), which is more precise that any other used previously, so if authors done the localization just visually it should be stated in the paper (as it means there could be more of nucleoli in karyotype and also some that are assumed to be nucleoli (like in arm F) might actually be just puffs).

- corrected. Visually we identified nucleoli (or puffs) in the arms D, F and G, the typical for Ch. riihimakiensis group

33)  Row 230: … mapping of chromosomal arms…

- corrected.

34)  Row 233: what morphological characteristics? This should be clarified.

- corrected.

35)  Row 251: better to say 'we observe" instead of "we note"

- corrected.

36)  Row 281: in this case I would add that the second sequence of Ch. riihimakiensis most probably doesn't belong to the species and it was mistake in species identification.

- in Discussion, line 361. “It could be a result of the misidentification by morphological characteristics of one of them or both [8,9]”.

37)  Row 290, figure 6: there is probably a mistake in naming as in pilicornis group there is Ch. heterodentatus. I assume it was supposed to be Ch. heteropilicornis? Please check this.

- corrected.

38)  Row 301: I would add here that identification of pupae and imago are also not 100% reliable as the problem of presence in GENEBANK of a lot of a lot of misidentified sequences appeared because imago was used for sequencing.

- corrected. – added “… pupae and imago morphology”.

39)  Row 313; remove "Chironomus"

- corrected.

40)  Row 315: I suggest to change it to: Between larvae from studied lakes we did not find Chironomus sp. Le1, only one hybrid with Ch. laetus.

- corrected.

41)  Row 319: As I mentioned above, the discussion of this should be changed completley as katyotypes are actually identical, it was description of it that was wrong.

- corrected.

42)  Row 326: remove Chironomus

- corrected.

43)  Row 334: excluding

- corrected.

44)  Row 347: Here I would also add a suggestion that the new cytocomplex (and the species) probably is very young as Ch. laetus not only is close to other species from Ch. riihimakiensis group but can form hybrids. Actually the finding of this hybrid is very unique as we indeed never saw such hybridization between species from different cytocomplexes (actually, mostly only sibling-species or very closely related species can form hybrids). So this finding is a gem of this work.

- corrected. “Our finding of a hybrid between Chironomus from different cytocomplexes is presented for the first time. This means that there is no complete reproductive isolation between two closely-related species. The new cytocomplex (and the species) probably is very young as Ch. laetus not only is close to other species from Ch. riihimakiensis group but can form hybrids”.

45)  Raw 349: no larvae of this species was found

- corrected.

46)  Row 365, 367, 369, 373: name of the sibling-species group should be written as "Ch. riihimakiensis", without ''''

- corrected.

47)  Row 377: suggests the presence

- corrected.

Round 2

Reviewer 3 Report

Manuscript Insects-1971820

General Comments. Overall, this manuscript is much improved. I still prefer to see greater reference to the importance of the habitat, but the authors are successful in showing that their work has significance beyond the sub-Arctic region.

Specific Comments

Lines 28-29: “First time in the lakes of the Yamal peninsula were found Chironomus laetus and the hybrid Ch. laetus x Ch. sp. Le1. T.” Change to “Initial research on the lakes of the Yamal peninsula resulted in the identification of Chironomus laetus and the hybrid Ch. laetus x Ch. sp. Le1. T.”

Line 88: Change “karyotype” to “the karyotype” or “its karyotype”.

Line 108: Change to “Chironomids are a very important component of aquatic ecosystems, and interest in them remains strong [7].”

Lines 109-110: Change “Molecular-genetic approaches are very popular last time and this requires high precision in identifying the species in different . . .” to “Molecular-genetic approaches are very useful in identifying the species in different . . .” If this is what you mean. The statement is not clear.

Lines 111-112: Your final sentence in the Introduction is “The study's goal is to learn more about the distribution, morphological, cytological, and molecular genetic characteristics of Ch. laetus and the hybrid of Ch. sp. Le1 x Ch. laetus.” I suggest making reference to the study sites as part of the distribution. For example, you could include “ . . . about the Arctic distribution”.

Lines 291-293: “During the rare investigation of rivers and lakes located outside of the Arctic Circle, every time it is possible to make interesting findings of chironomids (a new species, banding sequences etc.) [4,15,21,41].” Do you mean “just south of the Arctic Circle”? I suggest better defining what you mean by “outside” of the Arctic Circle.

Author Response

Thank you for your recommendations and corrections! Again.

General Comments. Overall, this manuscript is much improved. I still prefer to see greater reference to the importance of the habitat, but the authors are successful in showing that their work has significance beyond the sub-Arctic region.

- Thank you!

Specific Comments

Lines 28-29: “First time in the lakes of the Yamal peninsula were found Chironomus laetus and the hybrid Ch. laetus x Ch. sp. Le1. T.” Change to “Initial research on the lakes of the Yamal peninsula resulted in the identification of Chironomus laetus and the hybrid Ch. laetus x Ch. sp. Le1. T.”

- corrected.

Line 88: Change “karyotype” to “the karyotype” or “its karyotype”.

- corrected.

Line 108: Change to “Chironomids are a very important component of aquatic ecosystems, and interest in them remains strong [7].”

- corrected.

Lines 109-110: Change “Molecular-genetic approaches are very popular last time and this requires high precision in identifying the species in different . . .” to “Molecular-genetic approaches are very useful in identifying the species in different . . .” If this is what you mean. The statement is not clear.

- corrected.

“Molecular-genetic approaches are very useful in identifying the species in different stages of development, but this requires high accuracy in the first identification of species”.

Lines 111-112: Your final sentence in the Introduction is “The study’s goal is to learn more about the distribution, morphological, cytological, and molecular genetic characteristics of Ch. laetus and the hybrid of Ch. sp. Le1 x Ch. laetus.” I suggest making reference to the study sites as part of the distribution. For example, you could include “ . . . about the Arctic distribution”.

- corrected.

“The study's goal is to learn more about the Arctic distribution, morphological, cytolog-ical, and molecular genetic characteristics of Ch. laetus and the hybrid of Ch. sp. Le1 x Ch. laetus”.

Lines 291-293: “During the rare investigation of rivers and lakes located outside of the Arctic Circle, every time it is possible to make interesting findings of chironomids (a new species, banding sequences etc.) [4,15,21,41].” Do you mean “just south of the Arctic Circle”? I suggest better defining what you mean by “outside” of the Arctic Circle.

- corrected.

“During the rare investigation of rivers and lakes located in the Arctic zone, every time it is possible to make interesting findings of chironomids (a new species, banding sequences etc.) [4,15,21,41]”.
